# Gene Ontology Capsule GAN: an improved architecture for protein function prediction

Musadaq Mansoor, Mohammad Nauman, Hafeez Ur Rehman and Maryam Omar

National University of Computer and Emerging Sciences, Islamabad, Peshawar, KPK, Pakistan

## ABSTRACT

Proteins are the core of all functions pertaining to living things. They consist of an extended amino acid chain folding into a three-dimensional shape that dictates their behavior. Currently, convolutional neural networks (CNNs) have been pivotal in predicting protein functions based on protein sequences. While it is a technology crucial to the niche, the computation cost and translational invariance associated with CNN make it impossible to detect spatial hierarchies between complex and simpler objects. Therefore, this research utilizes capsule networks to capture spatial information as opposed to CNNs. Since capsule networks focus on hierarchical links, they have a lot of potential for solving structural biology challenges. In comparison to the standard CNNs, our results exhibit an improvement in accuracy. Gene Ontology Capsule GAN (GOCAPGAN) achieved an F1 score of 82.6%, a precision score of 90.4% and recall score of 76.1%.

## INTRODUCTION

Proteins play an integral role in a number of biological processes, performing many cellular functions (*Ashtiani et al., 2018*). Despite protein data being produced at an extremely high rate by different complex sequencing techniques, its functional understanding is yet to be discovered (*Rekapalli et al., 2012*; *Li et al., 2017*). Only about 1% of proteins have been explored and worked on experimentally, and they are manually annotated in the UniProt database (*Boutet et al., 2016*). *In-vitro* and *in-vivo* investigations can clarify and explain protein functions, but these methods have been shown to be time-consuming, expensive, and unable to keep up with the growing volume of protein data.

This encourages the development of a precise, efficient, and time-effective computational technique that can directly calculate protein functions from data. In this regard, a variety of approaches have been offered. In general, researchers build a pipeline that determines protein functions given protein sequences by performing the following steps: selection of a useful trait to encode input proteins, constructing datasets for experimenting and training purpose, selecting an appropriate algorithm, and evaluation of the performance. BLAST (*Altschul et al., 1990*) is a well-known computational technique which manually annotates the input sequences using the same functional sequences. As



Corresponding author
Musadaq Mansoor,
p177505@nu.edu.pk

popular as it is, BLAST has its shortcomings: (1) For a lot of input sequences, similar and functionally annotated sequences are hard to find; and (2) while some proteins have the same functions, they do not have sequence similarity. Hence, the results taken from methods like these that are based on homology are not always accurate and precise (*Pandey, Kumar & Steinbach, 2006*).

One option to overcome the drawbacks of other strategies is the extraction of relevant information from preserved subregions or input protein chain residues. *Das et al. (2015)*, proposed a domain-based technique for predicting protein functions, while *Wang et al. (2003)* presented, a motif-based function classifier for proteins. Finally, numerous approaches depend significantly on protein-protein interaction (PPI) information derived to properly compute and predict protein functions (*Jiang & McQuay, 2011*; *Peng et al., 2015*; *Chatterji et al., 2008*; *Hou, 2017*; *Nguyen, Gardiner & Cios, 2011*; *Rahmani, Blockeel & Bender, 2009*). The key concept backing these techniques is the idea that proteins with similar topological properties in PPI networks could also have similar functions (*Gligorijević, Barot & Bonneau, 2018*).

Moreover, quite a few protein function predictors require and use other types of data like making use of genomic context, (*Konc et al., 2013*; *Stawiski et al., 2000*; *Zhang, Freddolino & Zhang, 2017*; *Maghawry, Mostafa & Gharib, 2014*) exploiting protein structure, and (*Li, Tan & Ng, 2006*) consuming the knowledge of gene expression. We are now focusing on two types of predictors: sequence-based techniques (*Cai et al., 2003*; *Peng et al., 2014*) and PPI techniques (*Kulmanov, Khan & Hoehndorf, 2018*; *Rahmani, Blockeel & Bender, 2009*). PPI techniques that depend on data collected from these networks (*Kulmanov, Khan & Hoehndorf, 2018*; *Rahmani, Blockeel & Bender, 2009*), and sequence-based techniques that include using motifs, protein domains and residue-level information (*Cai et al., 2003*; *Peng et al., 2014*). Complementary data is often used by these strategies.

The proposed Gene Ontology Capsule GAN (GOCAPGAN) model is built on improving standard GANs to handle two essential issues: predicting functions based on sequence and annotating protein functions based on constrained categorized data. Therefore, different GAN variants were developed, primarily for picture synthesis challenges; whereas, just a few GAN variants are accessible for text generation problems. We used the current Wasserstein GAN (WGAN) model (*Arjovsky, Chintala & Bottou, 2017*) for our proposed version. We chose the WGAN model because of its high learning stability, ability to avoid mode collapse, and standard applicability for textual inputs, such as protein sequences in our situation. The use of GAN to tackle the issue of protein function prediction, as well as the originality of GOCAPGAN, are the study's main conceptual innovations. GAN is based on the use of unlabeled data, which is plentiful. Features are extracted from massive unlabeled datasets, which are utilized in the case of protein characterization. In order to generate protein sequences, the GAN is modified in the early phases. After generating sequences, the parameters of the GAN are tweaked in the second stage to predict protein functions based on the information gathered during sequence generation phase. Separate from the Uniprot database, the suggested prototype is tested on a dataset of proteins from *Homo sapiens*. In compared to previous techniques,

the results of the GOCAPGAN model show significant improvements in several evaluation measures.

It is evident that GANs are a fascinating and a field with rapid development and that delivers on the promise of generative models providing realistic samples in a variety of domains. GANs are an intelligent method of preparing a generating model by putting together a direct learning problem which has two associate models: generator, which is trained to produce new examples, and discriminator, which attempts to predict examples as fake (from outside the domain) or real (from the domain). In an adversarial zero-sum game, the two models are trained until the discriminator model is dodged almost half of the time, conveying that the generator model is producing proper samples.

GOCAPGAN framework utlizes anewly developed capsule network at the discriminator level, which sets it part from previous models. As CNN is translational invariant, they fail to capture relationship among features, whereas the recently introduced capsule network (*Sabour, Frosst & Hinton, 2017*; *Hinton, Sabour & Frosst, 2018*; *Hinton, Krizhevsky & Wang, 2011*) consists of capsules that are a group of neurons that encodes three-dimensional information of an object in addition to the probability of it being present. The capsule network is a new building element for deep learning that may be used to model hierarchical relationships within a neural network's internal knowledge representation. Contrary to CNN, information is encoded in vector form in capsules for storage of spatial data. For GOCAPGAN, the properties and features generated by the internal capsule layer explore the internal data distribution related to biological significance for enhanced outcome.

Capsule networks in current years have been used widely for object detection (*Lin et al., 2022*), automated email spam detection (*Samarthrao & Rohokale, 2022*), text classification (*Zhao et al., 2018*; *Zhao et al., 2019*; *Kim et al., 2020*), web blog content curation (*Khatter & Ahlawat, 2022*), fault diagnosis of rotating machinery (*Li et al., 2022*), identifying aggression and toxicity in comments (*Srivastava, Khurana & Tewari, 2018*), sentiment classification (*Chen & Qian, 2019*), biometric recognition system (*Jacob, 2019*) and simple classification hassles (*Lukic et al., 2019*; *Hilton et al., 2019*).

In the field of AI in biology and medicine, capsule networks have also been explored to inspect Munro's microabscess (*Pal et al., 2022*), brain tumor classification (*Afshar, Mohammadi & Plataniotis, 2018*; *Afshar, Plataniotis & Mohammadi, 2019*), pneumonia detection, and especially coronavirus disease 2019 (*Yang, Bao & Wang, 2022*; *Afshar et al., 2020*).

The current study is laid out as follows: "Literature review" looks at some studies that have been done on the problem of GO term prediction. "Methodology" goes over the implementation specifics in great detail. The important findings of this study are highlighted in "Results", which is followed by a discussion in "Discussion". Finally, in "Conclusion", the research is concluded with some future recommendations.

## LITERATURE REVIEW

The computational and experimental methods are two main ways of calculating protein functions. Experimental methods make use of biological experiments to confirm and

authenticate protein functions. One of the experimental methods is yeast two-hybrid (Y2H) used for recognizing protein functions. Y2H can examine an organism's entire genetic makeup for protein DNA interactions. Interactions in the worm, fly, and human (*Ghavidel, Cagney & Emili, 2005*) were recently discovered using Y2H. The disadvantage of this method is that it works on experiments, which necessitate adequate resources and laboratories. Another drawback of experimental approaches is that the time needed to characterize proteins cannot be predicted. Mass spectroscopy (MS) is a dynamic technique for examining protein interactions and predicting protein function. This method generates ions that may be detected using the mass to charge ratio, allowing for the identification of protein sequences (*Aebersold & Mann, 2003*). Like conventional methods, this procedure also has several drawbacks and limitations: it necessitates the use of qualified staff and appropriate equipment, and it is time-consuming. MS is very expensive, and protein complex purification limits protein characterization (*Shoemaker & Panchenko, 2007*). To predict protein activities, computational approaches use various protein information such as sequencing, structure, and other data available (*Lv, Ao & Zou, 2019*). These techniques may have drawbacks, but with reference to time and resource management these techniques are quite reasonable. Several methods, including machine learning algorithms and methodologies based on genomic context, homology and protein network, have proved successful in automatically predicting protein function. Machine learning advance models, such as deep learning, have been demonstrated to be more advanced than traditional machine learning models. Their superior performance is due to their capacity to assess incoming data automatically and more effectively represent non-linear patterns.

Protein function prediction and other bioinformatics applications have lately been done using deep-learning methods (*Deng et al., 2003*; *Nauman et al., 2019*). *Kulmanov, Khan & Hoehndorf (2017)* used DeepGO for function prediction. Deep learning was used to extract characteristics from protein interaction networks and sequences. One significant disadvantage of this method is that it necessitates a big amount of training data in order to make accurate predictions. It is also a computationally complicated model that consumes many resources (*Kulmanov, Khan & Hoehndorf, 2017*). *Kulmanov & Hoehndorf (2021)* also extended his work to DeepGOPlus. They created a unique technique for function prediction based solely on sequence. They combined a deep CNN model with sequence similarity predictions. Their CNN approach analyses the sequence for themes that predict protein activities and combines them with related protein functionalities (if available). The problem with this technique was that it worked better for similar sequences (*Kulmanov & Hoehndorf, 2021*). *Sureyya Rifaioglu et al. (2019)* used DEEPred for solving the function prediction problem. DEEPred was tuned and benchmarked utilizing three types of protein descriptors, training datasets of various sizes, and GO keywords from various levels. Electronically created GO annotations were also included in the training procedure to see how training with bigger but noisy data would affect performance (*Sureyya Rifaioglu et al., 2019*). *Gligorijević, Barot & Bonneau (2018)* used deep network fusion (deepNF) for the solution of function prediction. DeepNF is made up of multimodal deep auto encoders that extract proteins' important properties from a variety of networks with diverse interactions. To integrate STRING networks, DeepNF utilises high-level protein characteristics

constrained in a shared low-dimensional representation. For yeast/human STRING networks (*Gligorijević, Barot & Bonneau, 2018*), the results indicated that prior approaches had outperformed deepNF. DeepNF's main flaw is that it only uses the STRING network. This causes issues since functions expressed by a single protein are not taken into account. This is problematic because capabilities expressed by a single protein are not taken into account. DeepNF was found to be the best option for a few STRING networks. Deep learning methods have a major drawback in that they require a large amount of labelled data, whereas protein functions have a finite amount of labelled data. There is a large gap between protein sequences and function annotations. Furthermore, many GO keywords have only a few protein sequences, making deep learning algorithms difficult to forecast. GANs are used to isolate and extract patterns from unlabeled data, so they can perform well in the function prediction situation. Researchers have started utilizing GANs for producing biological data (*Gupta & Zou, 2018*). In our previous work, we also utilized GAN for protein function prediction (*Mansoor et al., 2022*).

Because of their affinity for hierarchical relationships, capsule networks have a lot of potential for solving structural biology challenges. DeepCap-Kcr (*Khanal et al., 2022*), a capsule network (CapsNet) based on a convolutional neural network (CNN) and long short-term memory (LSTM), was proposed as a deep learning model for robust prediction of Kcr sites on histone and nonhistone proteins (mammals). *de Jesus et al. (2018)* describes the implementation and application of a capsule network architecture to the classification of RAS protein family structures. HRAS and KRAS structures were successfully classified using a suggested capsule network trained on 2D and 3D structural encoding. In both of these studies, however, no biological data is synthesized for further study.

Given the success of the capsule network, *Upadhyay & Schrater (2018)* and *Jaiswal et al. (2018)* have investigated the capsule network with generative adversarial networks (GANs) and found promising results, but they have not investigated the capsule network with GANs for protein function prediction.

## METHODOLOGY

The GOCAPGAN model is based on the idea of modifying GOGAN (*Mansoor et al., 2022*) to solve the problem of predicting protein function from sparsely labelled data. The proposed paradigm can be divided into two stages. The first set of designs includes of the generator and discriminator architectures, which have been improved using residual blocks, with the last convolutional layer of a discriminator being replaced by a new and superior capsule network to record data in vector form. This modified model in this phase is prepared to produce protein sequences. In the second stage, after generating sequences, the altered GAN's parameters are utilized to forecast protein functions based on the knowledge that GAN gained during the sequence generation stage. We first present an introduction to classical GAN, for a more comprehensive insight, in the following subsections. Later, the proposed GOCAPGAN model's first stage is discussed, highlighting the important components of the GOCAPGAN model, namely the GOCAPGAN generator and GOCAPGAN discriminator. As capsule network plays the crucial role in the discriminator architecture, it has been discussed in detail prior to the explanation of

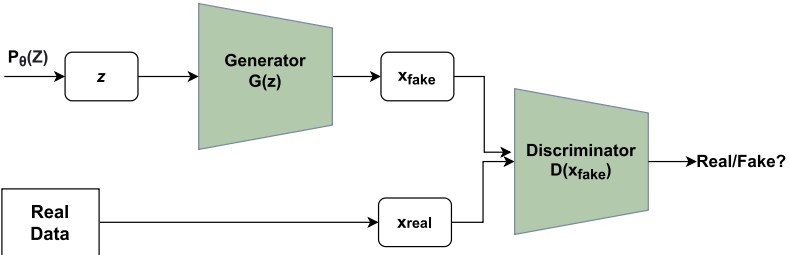

**Figure 1 Working of GAN.** Unpredictable noise $z$ from $p(z)$ is supplied to the generator, which produces data points. The $z$ denotes sample whereas $p(z)$ denotes probability distribution. The discriminator receives the data generated and assigns a value to it based on real data points or the generator. The discriminator then determines whether they are genuine or not.

discriminator architecture. Finally, the second stage is to be discussed, where the GOCAPGAN model parameters are utilized to forecast protein functions with the help of a multi-label classifier and transfer learning.

## Architecture of basic GAN

A novel framework had been given by Ian goodfellow that consisted of a system containing two major modules, namely generator and discriminator. Figure 1 depicts the basic idea of GAN where the generator utilizes noise vector $z$ as input creating novel data points and the discriminator functions as a classifier of the newly generated data points into a category of fake or real (*Goodfellow et al., 2014*).

The generator's main function is to produce realistic data points that are not classified by the discriminator as fake. Each run includes a back-propagation move that enhances the generator's parameters to facilitate the production of more realistic data points. If $x$ is an original data point, the discriminator returns a value $D(x)$ showing the likelihood. The goal is to increase the likelihood of correctly detecting real data points as opposed to created data points. Cross-entropy is used to calculate the loss: $plog(q)$ is a mathematical expression. The correct label for real data points is one, whereas the label for created data points is inverted. The main function of Discriminator is given in Eq. (1):

$$\max_D V(D) = \left[\mathbb{E}_{x \sim p_{data(x)}}[\log D(x)] + \mathbb{E}_{z \sim p_z(z)} \log(1 - D(G(z)))\right] \tag{1}$$

On the generator side, the generator's primary function is to create data points with the highest value of $D(x)$ in order to mislead the discriminator. Equation (2) provides the main function for the generator.

$$\max_G V(G) = \mathbb{E}_{z \sim p_z(z)}[\log(1 - D(G(z)))] \tag{2}$$

The goal functions of the generator and discriminator are learned simultaneously through interchanging gradient descent once they have been specified. The generator model parameters are fixed, and the discriminator undergoes a gradient descent iteration using both original and produced data points, after which the sides are swapped. The generator has been programmed for another cycle, and the discriminator has been

repaired. In alternate periods, both networks are trained until the generator delivers high-quality data points. Equation (3) depicts the GAN loss function:

$$\min_G \max_D V(D, G) = [\mathbb{E}_{x \sim p_{data(x)}}[\log D(x)] + \mathbb{E}_{z \sim p_z(z)} \log(1 - D(G(z)))] \tag{3}$$

$V(D, G)$ in Eq. (3), denotes entropy, which denotes real data points being supplied to the discriminator with the goal of increasing the entropy to one. While the second component of Eq. (3) indicates entropy, which shows created data points sent to discriminator, with the goal of reducing entropy to zero. Overall, the generator tries to decrease the objective function whereas the discriminator tries to maximize it.

GANs are frequently used to reduce divergences, although they are not always consistent with generator settings, which poses problems with GAN training. *Martin Arjovsky & Bottou (2017)* advised using Wasserstein-1 or Earth Mover's distance $W(q, p)$ to resolve this issue. The Wasserstein-1 or Earth Mover's distance is the amount of effort required to convert a q-distribution to a p-distribution with the least amount of effort. The Kantorovich-Rubinstein duality (*Villani, 2008*) is used by the WGAN objective function, which is provided by:

$$\min_G \max_{D \varepsilon \vec{D}} \mathbb{E}_{x \sim \mathbb{P}_r}[D(x)] - \mathbb{E}_{\vec{x} \sim \mathbb{P}_g}[D(\vec{x})] \tag{4}$$

We propose a new model called GOCAPGAN that is based on the notions of the classical GAN model discussed above. The proposed GOCAPGAN model is made up of two primary parts: the GOCAPGAN generator and the GOCAPGAN discriminator. The generator for the proposed model consists of multiple residual blocks in which each residual block contains dual convolutaional layers pursued by LeakuReLU, whereas for the discriminator, the internal structure of residual blocks is the same. However, instead of the last residual block, the capsule network is utilized. The GOCAPGAN model's general architecture is seen in Fig. 2.

## GOCAPGAN generator

The GOCAPGAN generator network produces protein sequences after the training. The GOCAPGAN generator's input size is specified as $(\Psi, 128)$, and $\Psi$ represents batch size. There were four distinct batch sizes tested: 16, 32, 48, and 64. Smaller batch sizes led to faster training, but provided lower accuracy. Due to restricted computing resources, the batch size for the suggested trials was set at 32. Once inputs are fed into the generator, it generates features or representations. The input latent vector is converted to low-level features by the generator using linear transformation. The generator network is built up of residual blocks rather than a traditional feed forward neural network. The GOCAPGAN generator is made up of six residual blocks. Each residual blocks uses two 1-D convolutional layers to learn information from given data. The activation function used is LeakyReLU. Gumbel Softmax outperforms softmax in terms of discrete text production (*Joo et al., 2020*). After experimenting with various sequence lengths, it was discovered that

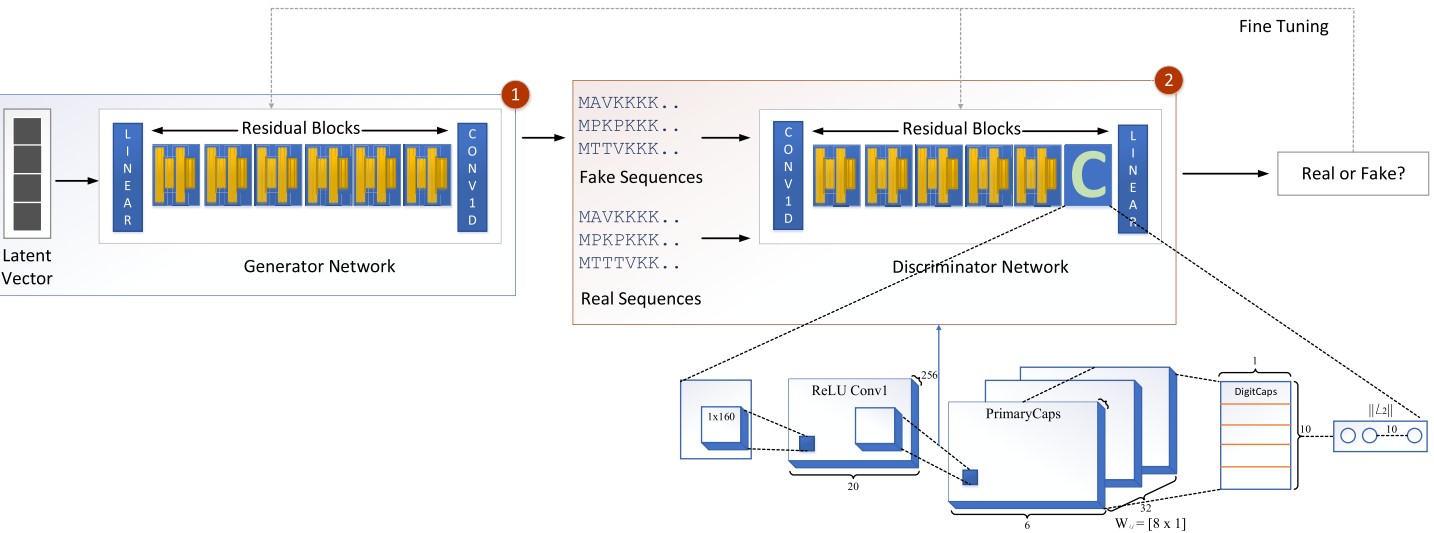

**Figure 2 GOCAPGAN GAN working.** The first step is passing the latent vector to the generator. Sequences are generated from the generator and passed to the discriminator. The second step is classification of generated sequence into real or bogus. The discriminator and generator model are readjusted so that the discriminator is unable to identify the generated sequences into original or fake.

sequence length 160 produced the best results. The total number of trainable parameters in the GOCAPGAN generator architecture is 18,447,894.

## Capsule network

The capsule network (CN) is an advanced neural network architecture conceptualized by Geoffrey E Hinton (*Sabour, Frosst & Hinton, 2017*). The goal of CN is to remedy some of CNN's shortcomings. In the past, CNN has been extensively studied in the fields of computer vision and other computer-assisted devices. They do, however, have several fundamental flaws and limits. In the following subsection, some limitations of CNN are discussed that motivates us to move towards CN.

### Translational invariant

Translational invariance is a property of CNNs. Consider an example to clarify what translational invariant implies, imagine that we have trained a model that can predict a presence of boat in a picture. Even if the identical image is translated to the right, CNN will still recognize it as a boat. However, because there is no method for CNN to predict translational property, this prediction ignores the extra information that the boat is moved to the right. Translational equivariance is required, indicating that the position of the object in the image should not be fixed in order for the CNN to detect it, but the CNN cannot identify the presence or position of one object related to others. Moreover, this results in difficulty identifying objects that hold special spatial relationship between features. In order to explain that, consider an example of a dissembled boat as depicted in Fig. 3. As CNN is looking for key features only, it will identify both as boat as spatial relationship between features is missing in case of CNN.

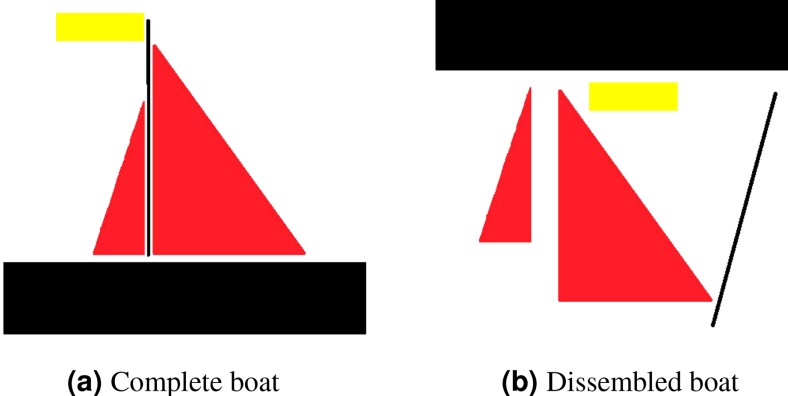

**(a)** Complete boat          **(b)** Dissembled boat

**Figure 3** For CNN, both (A) and (B) are boats, as mere presence of entities indicates object existence. However, for capsule network (A) is a boat whereas (B) is not considered to be a boat.

### Large data

In order to learn the features, CNN requires a lot of data to generalize the results. To overcome these constraints, the usage of CN is employed. Capsules are a group of neurons, where each neuron in a capsule represents various properties of a particular group of information. For example, if we consider four neurons each will be responsible for its own information like color, width, angle and height of particular information and the combination of all these four neurons is called as capsule. capsule dictates existence property, which means that there's a capsule in correspondence to each entity, which gives:

1. What is the likelihood that the information or entity exists.

2. Instantiation parameters of said entity.

The following are the main operations carried out within capsules:
Multiplication of the matrix of the input vectors with the weight matrix is calculated to encode the essential spatial link between low and high level features.

$$\hat{x}_{k|j} = W_{jk}x_j + B_k \tag{5}$$

The total of the weighted input vectors is used to select which higher-level capsule will receive the current capsule's output.

$$s_k = \sum_j c_{jk}\hat{x}_{k|j} \tag{6}$$

After that, the squash function is used to apply non-linearity. The squashing function reduces a vector's length to a maximum of one and a minimum of zero, while retaining its orientation.

$$v_k = squash(s_k) \tag{7}$$

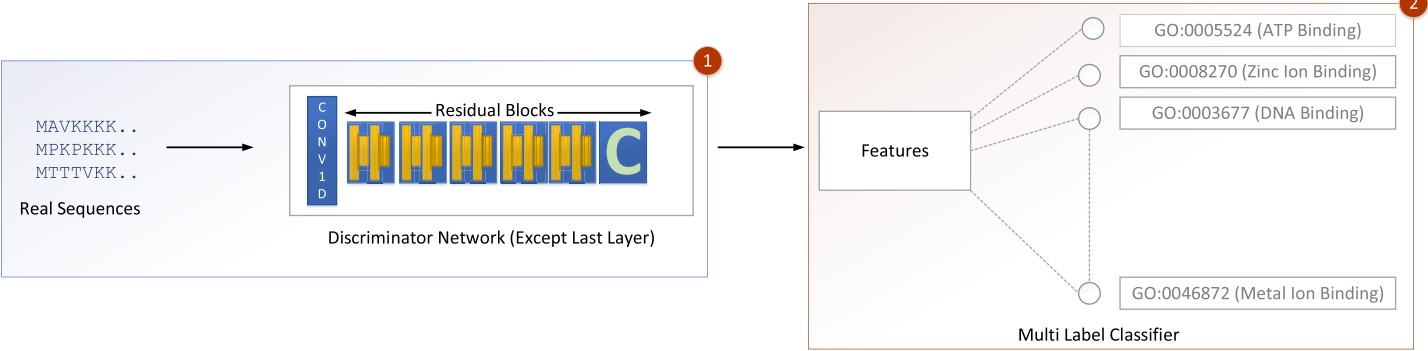

**Figure 4 GOCAPGAN transfer learning mechanism.** The first step deletes the last layer of the discriminator and save the initial weights. The upgraded discriminator then outputs features based on real protein sequences. The characteristics are then saved. Step 2 involves passing the stored features to a multi-label classifier for protein function projection.

## GOCAPGAN discriminator

The GOCAPGAN discriminator is divided into three sections to aid in the learning of how to discern between actual and fake proteins. To begin, low level features are obtained using a 1-D convolutional layer. The second part is a set of residual blocks for converting the data into a distinguishable state or set of values. The GOCAPGAN discriminator's third component is a linear layer that reports the probability of the input sequence being true or false and can be used to evaluate the discriminator's accuracy. The GOCAPGAN discriminator is given both the protein sequence generated by the GOCAPGAN generator and genuine protein sequences from the dataset. By traversing six residual blocks, the discriminator learns to distinguish between synthesised and actual protein sequences. The GOCAPGAN discriminator is made up of six residual blocks.

Figure 2 illustrates the proposed capsule network design incorporated in discriminator of GAN. The input data from a convolutional layer if fed to a capsule network that learns internal representations' essential properties. This layer's output is transmitted to the principal capsule layer, which creates a combination of the observed features.

The input is fed into a convolutional sub-layer in this layer, after which it is passed to a reshaped sub-layer, which prepares the data for the squash operation before being passed to the capsule layer. The dynamic routing operation takes three rounds in the capsule layer. The data is then transmitted to a length layer. Finally, to verify whether the input sequence is real or bogus, a linear transformation is applied. The GOCAPGAN discriminator architecture has a total of 4,029,697 trainable parameters. RMSprop (*Tieleman & Hinton, 2012*) was used as an optimizer, with alpha set to 0.99 and eps set to 1e−08. The rate of learning was set at 0.0001. Various optimization algorithms are available and have been tested; RMSprop delivered the best performance and accuracy on the stated dataset. Finally, the second stage is described in the following subsection, in which the GOCAPGAN model's parameters are used to predict protein functions using transfer learning and multi-label classifiers as shown in Fig. 4.

## Transfer learning

Transfer learning, also known as extracted features transferability, is a critical property of applying deep learning models to any problem. Transfer learning works by detaching the trained model's final layer, saving the weights of the previous layers, and then attaching a new final layer at the end. The features learned can be applied to a range of challenges if this update is useful for other sorts of classification.

Transfer learning was carried out in the GOCAPGAN model architecture in the following way: the discriminator contains all the traits that can be used to distinguish among the actual and forged protein sequences. As a result, only the GOCAPGAN discriminator was taken into account for transfer learning. This GOCAPGAN discriminator is now given genuine protein sequences without the last layer, and it generates features for them. These characteristics are then saved. The multi-label classifier receives the features obtained from this GOCAPGAN discriminator minus the last layer, as well as their functions or classifications.

## Multi-label classifier

The extracted features and classes are the two inputs to the multi-label classifier. The features extracted originate from putting genuine proteins through the upgraded GOCAPGAN discriminator, which is missing the last linear layer. The Gene Ontology (GO) class represents protein functions. The input is subsequently passed to the multi-label classifier's only dense layer. The dense layer outputs indicates the number of function projection. The dense layer uses a sigmoid activation algorithm. The binary crossentropy loss (*Vincent et al., 2010*) is used to calculate error, and it is given as:

$$J(\theta) = -\frac{1}{m}\left[\sum_{i=1}^{m} y^{(i)} log h_\theta(x^{(i)}) + (1 - y^{(i)}) log(1 - h_\theta(x^{(i)}))\right] \tag{8}$$

On the proposed model, many optimizers were tested but Adam provided the best performance and accuracy. Adam was used to train the GOCAPGAN multi-label classifier.

# RESULTS

A protein is associated in many processes whether they be biological, molecular or simple phenotypic, this essential piece of information is acquired from its function. It also clarifies how various molecules interact with one another. Several approaches for standardizing protein function concepts have been presented, and we choose the most frequently used, the Gene Ontology (GO). This model proposes the preparation and training of only all three elements of gene ontology. One of the best strategy for mass computational studies in GO since it has wide range of general adoption and consistency across species. The data and code files for the proposed GOCAPGAN model is available at: https://github.com/musadaqmansoor/gocapgan.

## Details of dataset

Proteins from *Homo sapiens* were used in the experiment. *The UniProt Consortium (2015)* database was used to acquire proteins. The system yielded a total of 72,945 proteins.

Tremble (tr) entries that had not been evaluated and swiss-port entries that had been reviewed were among the proteins (sp). Protein length is governed by the number of residues, which varies amongst proteins. In *Homo sapiens* species, the length ranges up to 34,358 residues. A few sequences exceed 2,000 residues in length, which is the highest residue length considered for computation. As a result, 70,956 proteins were employed in total. It's worth noting that our concept can be applied to different species without any changes. Longer sequences can be trained as well, however more computational resources and longer training time might be required.

## Target classes

*Homo sapiens* proteins were utilized in the suggested system. A conventional archive containing n all *Homo sapiens* protein sequences was used as the ground truth for the previously mentioned dataset. The suggested method is applicable to all conceivable *Homo sapiens* gene ontology classes. Because the GOCAPGAN model requires classes to have at least 16 protein sequences, there are 421 classes that are eligible to run the model. Twenty-five of these classes have been chosen for multi-label classification. The detail of these classes are given in Table 1. These classes were chosen based on the fact that they occur frequently.

## Setup for experiment

The suggested model is trained, tested, and validated using Google Colab as the standard system. CuDNN, Keras, Pytorch and Tensorflow libraries are used to implement the GOCAPGAN model in software.

## Preparation

### GOCAPGAN training

RMSProp is utilised as an optimizer for the GOCAPGAN model training, while Wasserstein loss is employed as an evaluation metric. Table 2 indicates values of hyper parameters for GOCAPGAN model training.

### GOCAPGAN classifier

Table 3 shows the various parameters and their values for multi-label classifier testing and training.

## Quantitative analysis

The system's performance was assessed *via* repeated k-fold cross validation. The number of splits (k) is three, and the number of repeats is five. The suggested model was evaluated using many performance indicators, including F1 score, recall, precision and hamming loss, which were stated as:

$$\text{precision} = \frac{\text{tp}}{\text{tp} + \text{fp}} \tag{9}$$

$$\text{recall} = \frac{\text{tp}}{\text{tp} + \text{fn}} \tag{10}$$

**Table 1 Classes for multi label classification.**

| Gene ontology | Description |
|---|---|
| GO:0046872 | Metal ion binding |
| GO:0005524 | ATP binding |
| GO:0003677 | DNA binding |
| GO:0008270 | Zinc ion binding |
| GO:0044822 | RNA binding |
| GO:0003700 | DNA-binding transcription factor activity |
| GO:0004930 | G protein-coupled receptor activity |
| GO:0042803 | Protein homodimerization activity |
| GO:0005509 | Calcium ion binding |
| GO:0004984 | Olfactory receptor activity |
| GO:0003723 | RNA binding |
| GO:0003682 | Chromatin binding |
| GO:0004674 | Protein serine/threonine kinase activity |
| GO:0043565 | Sequence-specific DNA binding |
| GO:0000166 | Nucleotide binding |
| GO:0005525 | GTP binding |
| GO:0000978 | RNA polymerase II cis-regulatory region sequence-specific DNA binding |
| GO:0042802 | Identical protein binding |
| GO:0019899 | Enzyme binding |
| GO:0019901 | Protein kinase binding |
| GO:0005102 | Signaling receptor binding |
| GO:0098641 | Cadherin binding involved in cell-cell adhesion |
| GO:0008134 | Transcription factor binding |
| GO:0031625 | Ubiquitin protein ligase binding |
| GO:0003924 | GTPase activity |

**Table 2 Parameters set for GOCAPGAN GAN training.**

| Parameter | Value |
|---|---|
| Batch size | 32 |
| Length of sequence | 160 |
| Epochs | 12 |
| Lambda | 10 |
| Noise | 128 |
| Rate of learning | 0.0001 |
| Optimizer | RMSprop |
| Loss function | Wasserstein loss |

**Table 3 Parameters set for GOCAPGAN multi-label classifier training.**

| Parameter | Value |
| --- | --- |
| Number of folds | 3 |
| Loss | Binary cross entropy |
| Number of repeats | 5 |
| Epochs | 40 |
| Optimizer | Adam |

**Table 4 Gene ontology classes results.**

| Method | Accuracy | Precision | Recall | F1 score |
| --- | --- | --- | --- | --- |
| GOCAPGAN | 84.1 | 77.4 | 93.2 | 84.5 |

**Table 5 Evaluation metrics of the GOCAPGAN model compared to GOGAN (*Mansoor et al., 2022*), DeepSeq (*Nauman et al., 2019*), and the BLAST (*Altschul et al., 1990*) method.**

| Method | Classes | Precision | Recall | F1 score | Hamming loss |
| --- | --- | --- | --- | --- | --- |
| GOCAPGAN | 25 | 0.904 | 0.761 | 0.826 | 0.085 |
| GOGAN | 10 | 0.852 | 0.625 | 0.721 | 0.095 |
| DeepSeq | 5 | 0.76 | 0.66 | 0.71 | 0.133 |
| BLAST | 5 | 0.46 | 0.64 | 0.53 | 0.387 |

$$F_1 score = 2 \times \frac{precision \times recall}{precision + recall} \tag{11}$$

$$HammingLoss = \frac{1}{NL} \sum_{l=1}^{L} \sum_{i=1}^{N} Y_{i,l} \oplus X_{l,i} \tag{12}$$

On 421 gene ontology classes, the performance of the GOCAPGAN model was calculated and reported in Table 4. The classes belonged to *Homo sapiens* only.

## Comparison with other techniques

The GOCAPGAN model, which was developed in this study, is compared to DeepSeq, BLAST and GOGAN. BLAST uses homology-based annotation transfer to predict protein function only based on sequence information. A local alignment algorithm known as BLAST is categorised as. BLAST searches for hits among protein sequences based on local region similarity. For proteins from the *Homo sapiens* species, precision, hamming loss, recall and F1 score of the GOCAPGAN, BLAST, GOGAN model, and DeepSeq are shown in the Table 5. As seen in Table 5, the GOCAPGAN model has approximately twice as many targeted classes as GOCAPGAN. In terms of precision and F1 score, the GOCAPGAN model outperforms DeepSeq, GOGAN and BLAST.

## DISCUSSION

The functional annotation of proteins is critical now that the genomes of various model species have been sequenced. In the current research, a deep learning based model GOCAPGAN is suggested that exploits generative adversarial networks along with capsule networks for synthesising protein sequences. As this process of synthetization enables our model to learn optimal features, these features are utilized in predicting protein function from sequences based on protein sequences. Unlike some methods of function prediction that are currently accessible, this model does not need custom-built attributes; instead, the architecture extracts information automatically from data sequences presented to the model.

GOCAPGAN uses a convolutional layer in conjunction with a capsule layer to capture more features. Capsules outperform standard CNN since it assimilates training and respective temporal association among various elements in one product. Experimentation results clearly indicate the usefulness of the suggested GOCAPGAN paradigm. For the time being, the model has only been tested and verified on the UniPort dataset, which is freely available. To further elaborate the importance of a capsule network, it was observed that each function is treated separately and independently in our method. In general, a protein's ability to perform one function does not exclude it from doing others. As a result, our approach predicts each protein's function without prejudice. Despite this, there are links between functions. Imagine a scenario in which you have highly related functions X and Y, and having function X increases your chances of getting function Y. Our method assumes that annotated proteins have detailed functional annotations and uses this information to predict functions for proteins that are not annotated. These annotated proteins could, in fact, have other activities that have yet to be found. With the passage of time and experimental research on protein function prediction, annotation of protein function may be on the path of further completion.

## CONCLUSION

In computational biology and bio-informatics, one of the major concern is determining functions of newly discovered proteins. Many conventional methods are still used to solve the gap between protein structure and function annotations, however these methods have a low accuracy. The current study proposes a novel new deep learning model for protein categorization built on the fusion of a capsule network and a GANs architecture. It shows how capsule networks can be applied to structural biology problems. To our knowledge, our team is the first in the field to use capsule networks in conjunction with GANs to build protein sequences that have also learned internal information. The results reveal that capsule networks outperform convolutional networks that have been around for a long time in terms of accuracy.

We intend to investigate further capsule network versions in the future, such as the Convolutional Fully-Connected Capsule Network (CFC-CapsNet) and Prediction-Tuning Capsule Network (PT-CapsNet). These developed architectures are unique and fast capsule networks, and they may provide an opportunity to identify additional qualities that could lead to higher assessment scores.

### Funding

The authors received no funding for this work.

### Competing Interests

The authors declare that they have no competing interests.

### Author Contributions

- Musadaq Mansoor conceived and designed the experiments, performed the experiments, analyzed the data, performed the computation work, prepared figures and/or tables, authored or reviewed drafts of the article, and approved the final draft.
- Mohammad Nauman performed the computation work, authored or reviewed drafts of the article, and approved the final draft.
- Hafeez Ur Rehman analyzed the data, authored or reviewed drafts of the article, and approved the final draft.
- Maryam Omar conceived and designed the experiments, performed the experiments, authored or reviewed drafts of the article, and approved the final draft.

### Data Availability

The data and code for the project is available on GitHub:

https://github.com/musadaqmansoor/gocapgan.

### Supplemental Information

Supplemental information for this article can be found online at http://dx.doi.org/10.7717/peerj-cs.1014#supplemental-information.

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
