# Peer review of "Gene Ontology Capsule GAN: an improved architecture for protein function prediction"

_PeerJ Computer Science, doi:10.7717/peerj-cs.1014_

## Round 0.1 · original submission · Minor Revisions

Your paper requires a number of revisions, please carefully revise and resubmit.

Please also make sure to get your paper proofread by a professional to increase its readability.

·

Basic reporting

The manuscript has been studied thoroughly and some recommendations for the authors are suggested , which are listed below:
1. In Figure 2, the author mentions a 9*9 input sequence in the detail design of the capsule network; nevertheless, the protein sequence is just one dimension. Please clarify this uncertainty.
2. Why did the authors use a capsule network at the GOCAPGAN discriminator level but not in the generator?
3. Figure 3 was not cited in the paper by the authors.
4. The authors mention in line 232 that they would consider an example of a disassembled boat, yet no visual picture of the boat is offered.

Experimental design

All necessary details regarding experiments are already provided in a manuscript.

Validity of the findings

The results reported by the author are aligned with the methodology discussed in a research article.

Additional comments

No Additional Comments

Reviewer 2 ·

Basic reporting

The paper is self-contained and is written very well.

Experimental design

The work is an extension of authors original work published earlier. The research questions are well defined and the proposed framework is described with sufficient details.

Validity of the findings

Numerous experiments are carried out to validate the findings. Publicly available datasets were used in the study.

Additional comments

The manuscript gives an improved concept of the author original work GOGAN. The authors introduced capsule network in GAN which lead to significant improvement in results. The improved technique was named GOCAPGAN. Introduction of the article contains all the important details and problem statement is given properly. The paper has already elaborated in detail that how utilization of capsule network can improve the results. Literature review is done in detail; methodology is written in such a way that it can easily be reproducible. Relevant tables and figures have been given. Conclusion and discussion properly relate the given solution with limitations in existing techniques. The grammar of the paper is in publishable form.
Some minor modifications are given below:
1- As mentioned in the paper GOCAPGAN is an extension of GOGAN, however there are many similarities between GOCAPGAN and GOGAN, so it is suggested to update some content of GOCAPGAN.
2- How the values of Table 2 and Table 3 were achieved?
3- In results, what is the difference between Table 4 and Table 5?

Reviewer 3 ·

Basic reporting

Authors have researched and written GOCAPGAN articles whose primary objective is to predict protein functions using GANs. They have used residual blocks, standard CNN, and capsule networks in the GANs. The improved technique works better in results and the number of classes. The reviews of each section are given below:
Abstract: The abstract of the submitted research paper is brief and displays all of the study article's material. The abstract is well organized, and it reports all of the findings.
Introduction: The introduction is done correctly. The introduction begins with a protein overview, an explanation of protein functions, and a demonstration of the sequence-structure gap. What is the difference between GAN (with CNN) and GAN (with Capsule), and how does it solves your prediction problem?

Literature Review: The writers conducted a study to compile background information and utilized current research articles to support the problem indicated in the introduction, however, include one or two recent literature reviews to complete it.

Experimental design

Methodology: The authors have described their technique in depth. The method gives information for the proposed model provided in figures and tables. The presented method in this work can be easily reproducible from the images and details shown. However, the authors did not cite Figure 3 in the article.

Validity of the findings

Results: The authors have supplied complete details regarding the dataset and hyperparameters. The authors performed rigorous testing and used different evaluation metrics and extensive testing. The source code of the implemented solution is available on GitHub. The authors need to explain why they did not include a hamming loss in Table 4.

Additional comments

Discussion: The problem statement in the introduction is linked to the results, and the findings support the recommended solution to the problem.

Conclusion: The conclusion gives a summary of the significant points throughout the article and supports the author’s claims
The paper structure is sound and written in an acceptable way. The quality of the English written in the research paper is satisfactory for publication.

---

## Round 0.2 · accepted · Accept

Thanks for your contribution to PeerJ Computer Science.

·

Basic reporting

The article is written in English and is clear, unambiguous, technically correct text. The article conforms to professional standards of courtesy and expression.

The article has sufficient introduction and background to demonstrate how the work fits into the broader field of knowledge. Relevant prior literature is appropriately referenced.

Experimental design

Original primary research within Aims and Scope of the journal.

Validity of the findings

All underlying data have been provided; they are robust, statistically sound, & controlled.

Additional comments

All suggested changes are incorporated in updated article. The article meets the PeerJ criteria and should be accepted as is.

Reviewer 2 ·

Basic reporting

The authors have addressed all of my concerns. I have no further comments.

Experimental design

Research questions are well defined and rigorous investigation has been performed.

Validity of the findings

Results are validated on publicly available datasets. Code and datasets are provided on the GitHub.

Reviewer 3 ·

Basic reporting

I am happy with the comments of round 1 being incorporated. The manuscript is relevant to the journal scope and has significant contributions—no further suggestion from my side. Therefore, I propose to accept it.

Experimental design

Comments of round 1 are incorporated—no further changes are required from my side.

Validity of the findings

Comments of round 1 are incorporated—no further changes are required from my side.

Additional comments

Comments of round 1 are incorporated—no further changes are required from my side.